# Anti-PD-L1 Immunotherapy of Chronic Virus Infection Improves Virus Control without Augmenting Tissue Damage by Fibrosis

**DOI:** 10.3390/v16050799

**Published:** 2024-05-17

**Authors:** Valentina Casella, Paula Cebollada Rica, Jordi Argilaguet, Enric Vidal, María González-Cao, Roberto Güerri-Fernandez, Gennady Bocharov, Andreas Meyerhans

**Affiliations:** 1Infection Biology Laboratory, Department of Medicine and Life Sciences (MELIS), Universitat Pompeu Fabra, 08003 Barcelona, Spain; paula.cebollada@upf.edu; 2Institute of Agrifood Research and Technology (IRTA), Centre de Recerca en Sanitat Animal (CReSA), 08193 Barcelona, Spain; jordi.argilaguet@irta.cat (J.A.); enric.vidal@irta.cat (E.V.); 3Unitat Mixta d’Investigació IRTA-UAB en Sanitat Animal, Centre de Recerca en Sanitat Animal (CReSA), Campus de la Universitat Autònoma de Barcelona (UAB), 08193 Barcelona, Spain; 4WOAH Collaborating Centre for Emerging and Re-Emerging Pig Diseases in Europe, IRTA-CReSA, 08193 Barcelona, Spain; 5Instituto Oncologico Dr. Rosell, Hospital Quiron-Dexeus Barcelona, 08028 Barcelona, Spain; mgocao@gmail.com; 6Infectious Diseases Unit, Hospital del Mar, Institute of Medical Research (IMIM), 08003 Barcelona, Spain; robertoguerri@gmail.com; 7Marchuk Institute of Numerical Mathematics, Russian Academy of Sciences, 119991 Moscow, Russia; gbocharov@gmail.com; 8Institute of Computer Science and Mathematical Modeling, Sechenov First Moscow State Medical University, 119635 Moscow, Russia; 9Institució Catalana de Recerca i Estudis Avançats (ICREA), 08010 Barcelona, Spain

**Keywords:** immunotherapy, anti-PD-L1, fibrosis, chronic virus infection, LCMV

## Abstract

Immunotherapy with checkpoint inhibitors, albeit commonly used against tumors, is still at its infancy against chronic virus infections. It relies on the reinvigoration of exhausted T lymphocytes to eliminate virus-infected cells. Since T cell exhaustion is a physiological process to reduce immunopathology, the reinvigoration of these cells might be associated with an augmentation of pathological changes. To test this possibility, we here analyzed in the model system of chronic lymphocytic choriomeningitis virus (LCMV)-infected mice whether treatment with the checkpoint inhibitor anti-PD-L1 antibody would increase CD8 T cell-dependent fibrosis. We show that pre-existing spleen fibrosis did not worsen under conditions that increase CD8 T cell functionality and reduce virus loads suggesting that the CD8 T cell functionality increase remained below its pathogenicity threshold. These promising findings should further encourage immunotherapeutic trials against chronic virus infections.

## 1. Introduction

Fibrosis, or scarring, is the result of a reparative response to inflammatory insults or tissue damage [1]. It is characterized by the deposition of collagen and other extracellular matrix proteins, and a major cause of tissue and organ dysfunction. For the United States alone, fibrosis was estimated to be involved in up to 45% of all deaths [2].

Virus infections can trigger immunopathologies and fibrosis in infected tissue due to the induction of inflammatory antiviral immune responses [3]. A well-characterized example for acute infections is the development of pulmonary fibrosis after SARS-CoV-2 infection [4]. It is the most significant post-acute sequelae of COVID-19, and leads to long-term impairment of pulmonary function. Examples for fibrosis after chronic infections are the induction of lymphatic tissue and liver fibrosis in HIV-1 infection [5,6] and chronic hepatitis B or C infection, respectively. For HIV-infected individuals, lymphatic tissue fibrosis can lead to a reduction in immune responsiveness towards vaccines [7] and increased susceptibility to other infections [8]. HBV- and HCV-infected individuals may develop liver fibrosis with high risks of progressing towards metabolic pathologies and liver cancer [9]. Therefore, therapeutic strategies should always be carefully evaluated to avoid worsening of tissue fibrosis.

Using the acute and chronic lymphocytic choriomeningitis virus (LCMV) infection model of mice, we have previously shown that spleen fibrosis occurs after both types of infections, and that it is mediated by antiviral CD8 T cells in an interferon type I (IFN-I)-dependent manner [10]. In chronic infections and cancers, these cytotoxic T cells show an exhausted phenotype with reduced effector functions [11]. This phenotype is part of the system’s response to limit immunopathology [12]. Current immunotherapeutic strategies utilise checkpoint inhibitors like anti-PD-1 or anti-PD-L1 antibodies to invigorate exhausted CD8 T cells [13] and thus may not only provide a therapeutic benefit but also increase fibrosis. In this work we tested whether the immunotherapy-mediated restoration of CD8 T cell functionality in chronically infected mice would lead to an increase in lymphoid tissue fibrosis as a consequence of the increased killing of infected cells. For this, we administered three different regimens of anti-PDL1 antibodies and analyzed possible correlations between CD8 T cell responses and spleen fibrosis. We show that all the immunotherapeutic regimens successfully reactivated T cell function without worsening the preexisting fibrosis.

## 2. Results

The potent antiviral CD8 T cell response that is induced after an experimental LCMV-Doc infection of mice causes spleen fibrosis [10]. Fibrosis is detectable already 2 weeks post-infection (p.i.) and still persists at week 5 p.i. [10,14]. Since the augmentation of CD8 T cell function during chronic virus infections or cancers are a major component of current immunotherapy trials, it is conceivable that these procedures concurrently induce or worsen fibrosis as a side effect. To address this possibility, we used chronic LCMV infection as a model system and evaluated the impact of anti-PD-L1 (aPDL1) antibody immunotherapy on virus load, T cell function and spleen fibrosis. For this, C57BL/6j mice were chronically infected with 2 × 10^6^ plaque forming units (PFU) of LCMV-Docile (LCMV-Doc) and subjected to 3 different regimens of anti-PD-L1 antibody treatment. They varied in the number (3 or 5) and timing (early or late) of anti-PD-L1 administration (Figure 1a). CD8 T cell function and exhaustion were analysed one day after the last administration of aPDL1 (Figure 1b,c) by interferon-γ (IFNγ) intracellular cytokine staining (ICS) and surface expression of PD1 and Tim3, respectively. Anti-PDL1 antibody treatment also increases absolute numbers of LCMV-specific CD8 T cells, in particular progenitor 2 exhausted (T_PEX_2) and intermediate exhausted (T_EX_INT) T cells, as previously shown [15]. Splenic virus loads (Figure 1d) and fibrosis scores (Figure 1e,f) were determined at days 30, 35 and 42 p.i.. Both, the short and the long late anti-PD-L1 treatment improved the functionality of LCMV-specific CD8 T cells compared to the corresponding untreated controls at days 35 and 42 p.i. from approximately 1.8% to 4% and from 0% to 4%, respectively (Figure 1b). This improved T cell functionality seems to correlate with the decrease in splenic virus loads (Figure 1d). On the contrary, early anti-PD-L1 treatment did not further improve the frequency of LCMV-specific IFNγ-producing CD8 T cells (Figure 1b), nor reduced virus loads in spleen (Figure 1d) when compared to the untreated controls. Thus, immunotherapy with anti-PD-L1 antibody is not effective around the onset of T cell exhaustion when chronicity of the infection is established [14], however it is effective at later time points of the chronic infection phase and reduces virus loads.

Interestingly, all three anti-PD-L1 regimens restored CD8 T cell functionality to similar frequencies (around 4% of activated CD8 T cells; (Figure 1b)) suggesting that this is the achievable treatment threshold. Furthermore, the frequency of exhausted T cells, defined as CD44 + PD1 + Tim3 + CD8 T cells, was not reversed by any of the three anti-PD-L1 regimens during the duration of the treatment (Figure 1c). This is concordant with our previous observation that the functionality increase after anti-PD-L1 is mainly observed in the exhausted T cell subset [15].

To then evaluate the potential impact of anti-PD-L1 immunotherapy on spleen fibrosis in chronic LCMV infection, spleens from treated and untreated animals were fixed in formalin, embedded in paraffin and cut with a microtome. Collagen fibers indicative of tissue fibrosis were visualized by Masson’s Trichrome staining. All chronic LCMV-infected mice, either anti-PD-L1 treated or untreated, displayed a similar degree of splenic fibrosis which was absent in naive mice (Figure 1e,f).

To test for (i) a direct correlation between improved T cell functionality and splenic virus load control and (ii) the maintenance of the LCMV-specific CD8 T cell response after immunotherapy, chronic infected mice were treated with anti-PD-L1 (Figure 2a) and analyzed 30 days after treatment (day 60 post-infection) for specific CD8 T cell responses (Figure 2b), virus loads (Figure 2c) and fibrosis score (Figure 2d). Anti-PD-L1 resulted in an increased T cell functionality even 1 month after treatment termination and was linked to virus control. The fibrosis score was not affected by the therapy. Thus, anti-PD-L1 immunotherapy-mediated reactivation of the exhausted CD8 T cell response in chronic LCMV-infected mice is sufficient to improve virus control without increasing the associated tissue damage and subsequent fibrosis.

## 3. Discussion

Our study reveals several interesting elements relevant for the immunotherapy of chronic virus infections. First, the time point of immunotherapy initiation with checkpoints inhibitors matters. Only when exhaustion is fully established, T cell invigoration and a therapeutic antiviral effect becomes evident. This is in line with the observation that exhaustion is a sequential process with distinct differentiation states [16]. Second, from the total pool of exhausted activated cells (Figure 1c), only a fraction can be functionally reactivated with a defined antigen specificity (Figure 1b). Since this reactivation was independent of the type of anti-PD-L1 antibody regimen, this level may represent the achievable threshold of the infected host organism. Third, anti-PD-L1-mediated CD8 T cell reactivation during the chronic LCMV infection phase was below the level that would worsen the pre-existing tissue fibrosis. This observation supports the feasibility of this immunotherapeutic approach to control virus infections even in fibrosis-bearing patients like those infected with HIV. Fourth, the anti-PD-L1-mediated CD8 T cell reactivation was maintained even 1 month after stopping treatment suggesting longer term benefits that may go beyond antibody half-lives. Finally, the anti-PD-L1 antibody treatment period of 15 days in mice would correspond to over a 1-year treatment period in humans when considering mice to human lifespan correlations thus covering a physiologically relevant and immunologically informative time frame [17].

The immunopathology that is associated with virus elimination during acute infections is considered as a ‘recovery fee’. During chronic infections, virus-host interactions show various degrees of immunopathology that can manifest as alterations in organs’ structural organization [18], e.g., organ fibrosis. Importantly, whereas immune responses to acute infections are always accompanied by inflammatory reactions, persistent virus infections differ with respect to the level of chronic inflammation which is, for example, high in HIV-1 [19] as a result of massive bacterial antigen dislocation to the systemic circulation, but less prominent in LCMV infection [20]. Interestingly, inflammation itself that underlines tissue injury is not necessary linked to fibrosis. Rather, it is the level and the kinetics of the inflammatory response over time which determines whether tissue repair will lead to progressive fibrosis or end in efficient repair [21]. Within a multi-component inflammatory response, TGF-β is the key factor for fibrosis induction [22]. Our study now shows that anti-PD-L1 works within a narrow window of immune networks without engaging fibrotic processes. Hence the control of viral load in chronic infections may be achieved without touching the fibrotic loop and thus without a ‘functional recovery fee’.

As of today, immunotherapy with checkpoint inhibitors is not part of the clinical practice against chronic virus infections. Their therapy potential in reducing virus loads has been addressed in several animal model studies [23,24], in a theoretical modelling approach [25] but only few human studies that mainly involved chronically infected individuals with tumors [26,27,28]. In contrast, checkpoint inhibitors are well established in tumor treatment for which they have become a standard therapy component for a third of tumor pathologies. Consequently, therapy side effects like fibrosis induction are mainly discussed in the latter context. In animal models of idiopathic pulmonary fibrosis, PD-1/PD-L1 blockade has been shown to be therapeutic and can reduce fibrosis that occurs due to hyperactivation of myofibroblasts induced by TGF-β [29,30]. In lung cancer patients with previous fibrosis, treatment with anti PD-1/PD-L1 antibodies increases the risk of immune mediated pneumonitis and thus may be detrimental [31]. Furthermore, it is well known that highly fibrotic tumors limit T lymphocyte entry into the tumor microenvironment (T cell excluded tumors) and thus are resistant to anti-PD-1/PD-L1 antibody therapy. When immune-mediated adverse effects occur, if they persist over time and become chronic, they can trigger fibrosis. Moreover, although uncommon, some cases of IgG4-related retroperitoneal fibrosis mediated by B cell activation have been reported during cancer immunotherapy [32]. Together these data suggest that the side effects of checkpoint inhibitor therapies including fibrosis strongly depend on the specific pathophysiological context in which they are used and should be carefully monitored to guarantee patient benefit.

In summary, this study emphasizes the T cell functionality increase after different anti-PD-L1 immunotherapy regimens during a pathogenic chronic virus infection that leads to improved virus control without augmenting tissue damage by fibrosis. It emphasizes the need to carefully monitor possible signs of exaggerated inflammatory responses and fibrosis when intending to better control human infections such as those with HIV. Limitations of this study are the sole use of a virus infection model system in mice, the lack of long-term animal follow-up beyond 2 months of infection, and the use of only anti-PD-L1 as checkpoint inhibitor. Nonetheless, we provide encouraging findings to go ahead with immunotherapy trials against chronic virus infections in humans.

## 4. Material and Methods

### 4.1. Mice

Male C57BL/6J mice (RRID:IMSR_JAX:000664) aged 6 weeks were purchased from Charles River Laboratories. All mice were bred and maintained under specific pathogen-free conditions at in-house facilities. All animal procedures were conducted according to the guidelines from Generalitat de Catalunya and approved by the ethical committees for animal experimentation at Parc de Recerca Biomèdica de Barcelona (CEEA-PRBB, Barcelona, Spain).

### 4.2. Virus Infections

LCMV strain Docile (LCMV_Doc_) was used for mouse infections. The virus was grown in L929 cells, titrated using focus forming assay on MC57 cells and stored according to previously published methods [14]. Mice were infected by intraperitoneal (i.p.) injection using a high dose (2 × 10^6^ plaque-forming units (pfu)) of LCMV_DOC_ to induce a chronic infection.

### 4.3. In Vivo Treatments with Anti-PD-L1

Mice were administered i.p. with 200 μg of anti-PD-L1-specific mAb (10F.9G2, BioXCell, Lebanon, NH, USA) in three to five shots every three days. In particular, early aPDL1-treated mice were given three shots at days 11, 14 and 17 p.i. Short late aPDL1-treated mice were given three shots at days 22, 25 and 28 p.i. and long late aPDL1-treated mice were given five shots at days 19, 22, 25, 28 and 31 p.i. Mice analysed at day 60 post-infection, were given four shots of aPDL1 at days 22, 25, 28 and 31 p.i. As a control, physiological serum was administered.

### 4.4. Flow Cytometry

Spleens were mechanically disrupted, processed to obtain single-cell suspension and stained as previously described [14]. Equal number of splenocytes were stained for viability with Live/Dead Fixable Violet Cell Stain (ThermoFisher Scientific, Waltham, MA, USA). After washing, cells were stained with fluorochrome-labeled antibodies targeting surface or intracellular proteins. For detection of IFNγ-producing T cells, intracellular cytokine staining was performed after 5 h ex vivo stimulation with GP_33_(1 μg/mL) in the presence of Brefeldin A (BFA, Sigma, Burlington, MA, USA). After surface antibody staining, cells were fixed with 2% Formaldehyde, permeabilized in Perm/Wash buffer (PBS 1% FCS, NaN_3_ 0.1%, Saponin 0.1%) and then stained with intracellular antibodies for IFNγ. Flow cytometry samples were acquired using the Aurora (Cytek, Fremont, CA, USA) analyzer. FACS data were analyzed using FlowJo 10.1 software (Tree Star Inc., Ashland, OR, USA)

### 4.5. Virus Load Quantification

Viral titers from serum and spleens of infected mice were determined by focus forming assay as previously described [14]. Respectively, blood was collected before sacrifice and centrifuged at 5000× *g* (RCF) for 10′ to take the serum, spleens were harvested and frozen at −80 °C. Tissue was mechanically disrupted and resuspended in 500 μL of DMEM 2%FBS. For both serum and disrupted spleen, one in ten-fold dilutions were prepared and overlaid onto MC57 cell monolayers in 24-well plates. After 48 h of incubation at 37 °C 5% CO_2_, staining was performed using monoclonal rat anti-LCMV antibody (VL-4) for 1 h, followed by Peroxidase Anti-Rat IgG Polyclonal Ab (Jackson ImmunoResearch, West Grove, PA, USA). Plaques were visualized using DAB Peroxidase substrate kit (Vector Laboratories, Newark, CA, USA).

### 4.6. Histological Staining

Spleen samples were fixed overnight with 4% buffered formaldehyde and paraffin-embedded. For semi quantification of fibrosis, Masson’s Trichrome was performed on 3 µm thick spleen cuts that revealed collagen fibers in blue. A semiquantitative fibrosis score from 0 to 10 was defined. A score of 0 represents normality, while 10 would correspond to a spleen in which the parenchyma has totally been replaced by connective tissue.

### 4.7. Quantification and Statistical Analysis

Graphs were compiled and statistical analyses were performed with Prism (GraphPad software v9.0). Statistical significance was evaluated with the unpaired t-test when comparing two groups and one-way ANOVA multiple comparison test when comparing more than two groups. Non-significant differences were indicated as “ns”. *p*-values below 0.05 were considered significant and were indicated by asterisks: * *p* < 0.05; ** *p* < 0.01; *** *p* < 0.001.

## Figures and Tables

**Figure 1 viruses-16-00799-f001:**
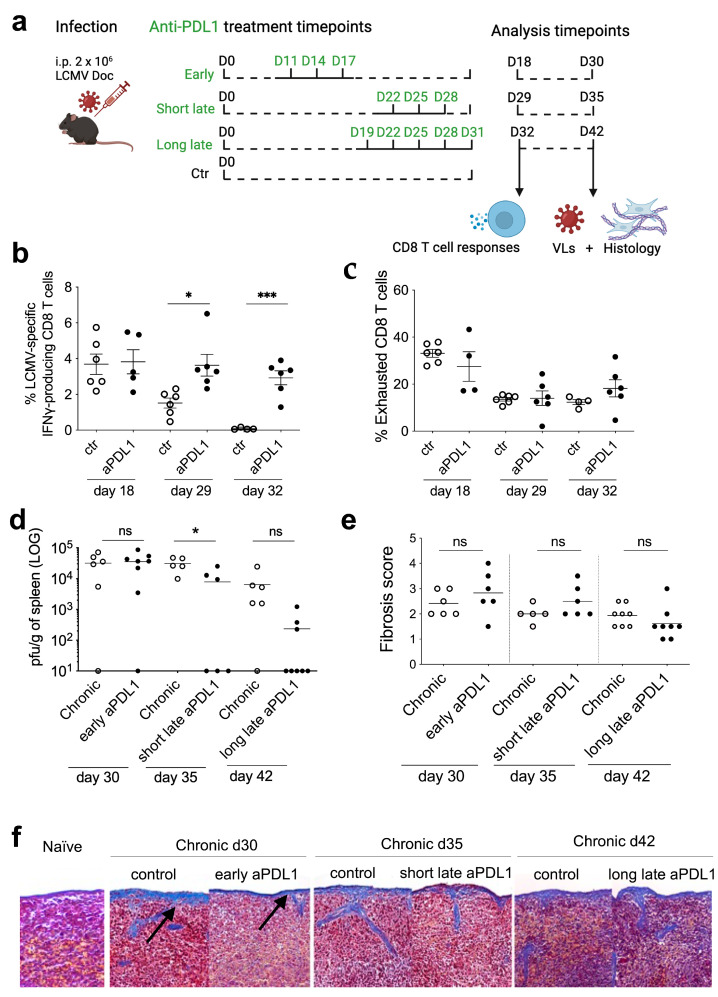
Immunotherapy with anti-PD-L1 improves CD8 T cell-mediated virus control without worsening spleen fibrosis in LCMV chronic infected mice. (**a**) Schematic representation of three different regimens of anti-PD-L1 (aPDL1) antibody treatment in chronic LCMV-infected mice. Untreated chronic infected mice were used as controls (ctr) and sampled at the same time points as the anti-PD-L1-treated mice. (**b**,**c**) Frequencies of LCMV gp33-responsive IFNγ-producing CD8+ T cells (**b**) and of Tim3+PD1+ exhausted CD8+ T cells (**c**) on days 30, 35 and 42 post-infection, after the three different aPDL1 regimens. (**d**) Viral loads in spleen from days 30, 35 and 42 post-infection, after the aPD-L1 regimens. (**e**,**f**) Spleen fibrosis score (**e**) and representative images of Masson’s Trichrome stained spleen sections (**f**) used to score the fibrosis in each group on days 30, 35 and 42 post-infection; the black arrows indicate deposits of collagen fibers (in blue—Masson trichrome stain) characteristic of fibrosis. Naive mouse spleen is stained as control. Data shown are the mean of *n* = 5 to 8 mice. * *p* ≤ 0.05; *** *p* ≤ 0.001; ns = not significant (unpaired two-tailed *t*-test).

**Figure 2 viruses-16-00799-f002:**
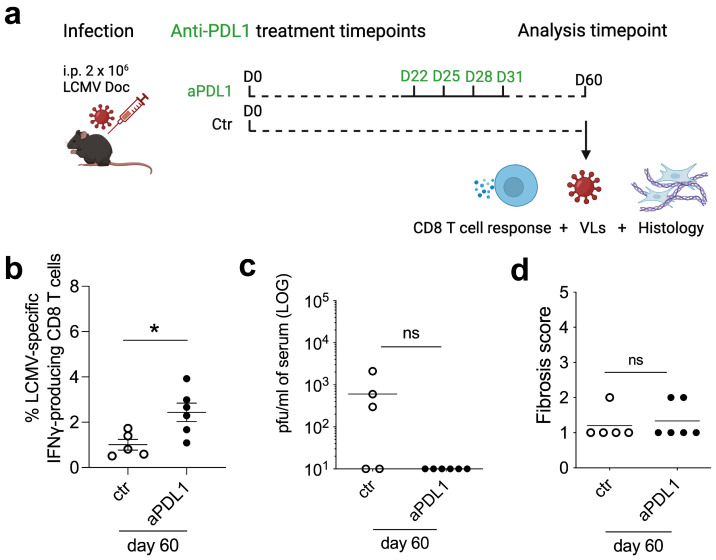
Persistent increase of CD8 T cell-mediated virus control after anti-PD-L1 antibody treatment of chronic LCMV-infected mice. (**a**) Schematic representation of anti-PD-L1 (aPDL1) antibody treatment in chronic LCMV-infected mice. Untreated chronic infected mice were used as controls (ctr) and sampled at the same time point as the anti-PD-L1-treated mice. (**b**) Frequencies of LCMV gp33-responsive IFNγ-producing CD8+ T cells at day 60 post-infection and 30 days after the last aPDL1 treatment. (**c**) Viral loads in serum from day 60 post-infection, after aPDL1 treatment. (**d**) Spleen fibrosis score from day 60 post-infection. Data shown are the mean of n = 5 to 6 mice. * *p* ≤ 0.05; ns = not significant (unpaired two-tailed *t*-test).

## Data Availability

The data presented in this study are available upon request from the corresponding authors.

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
