# Peer review of "Anti-PD-L1 Immunotherapy of Chronic Virus Infection Improves Virus Control without Augmenting Tissue Damage by Fibrosis"

_viruses, 2024, doi:10.3390/v16050799_

Round 1
Reviewer 1 Report
Comments and Suggestions for Authors
I have read and reviewed the manuscript entitled “Anti-PDL1 checkpoint inhibitor treatment during a chronic virus infection acts below a tissue pathogenicity threshold” authored by Casella, V., et al ., which was submitted to Viruses for consideration. This brief communication is an extension upon the more comprehensive work detailed in reference 10 in this manuscript. The hypothesis is that the usage of check point inhibitors will enhance LCMV clearance by expanding virus-specific CD8T cell-responses in a model of LCMV chronicity, which is documented, but expected. The second part is that immune re-activation would enhance spleen fibrosis, which it did not. Without further exploration, the conclusion the title implies can not be supported. The minimal amount of experiments described in the paper and the hard to compare experimental groups, preclude me to recommend its publication in Viruses.
Major points
- Only two experimental groups (early and short late) are directly comparable. The “long late”experimental group contains two additional doses and a different schedule. Therefore, the support for the therapeutic timing can not be supported.
- Fibrosis seems time-dependent, mainly in the treated groups. Authors should expand this analysis.
Minor points
- Figures 1b and 1c should be plotted using total number of CD8 lymphocytes.
- Lines 63 to 67 belong to the introduction.
- A figure with viremia needs to be included.
Reviewer 2 Report
Comments and Suggestions for Authors
This study investigated whether immunotherapy to reinvigorate exhausted T cells during chronic LCMV infection in mice would worsen pre-existing spleen fibrosis. Mice were treated with anti-PDL1 at different timepoints during chronic LCMV infection. All anti-PDL1 treatments improved T cell function and reduced viral loads but did not increase fibrosis scores compared to untreated mice. The results show that anti-PDL1 immunotherapy can enhance virus control without exacerbating tissue damage and fibrosis. The findings support feasibility of immunotherapy against chronic infections like HIV without worsening fibrosis.
Here are some suggestions to improve the quality of the current presentation:
- Avoid verbosity - The summary could be more concise by removing unnecessary words like "investigated whether" and using simpler phrasing.
- Improve clarity - Using more specific language like "anti-PDL1 antibody treatment" instead of just "anti-PDL1" would make it clearer for readers unfamiliar with the terminology.
- Highlight key findings - Putting the most important result (that anti-PDL1 did not worsen fibrosis) earlier in the summary would highlight the key takeaway.
- Add implications -Mentioning the potential clinical relevance of using this approach safely in diseases like HIV would underscore the impact of the findings.
none.
Reviewer 3 Report
Comments and Suggestions for Authors
This is a well-written article that requires some clarifications. My comments are attached.

Author Response
Dear Editor,
Thank you for handling our manuscript now entitled "Anti-PDL1 immunotherapy of chronic virus infection improves virus control without augmenting tissue damage by fibrosis" (viruses-2855839) and for giving us the opportunity to revise it according to the reviewer´s comments.
We have now addressed the issues brought up by reviewer 3 and modified the manuscript accordingly. In our response letter to the reviewer´s comments, you will find a point-by-point response to each issue brought up. For clarity, we have highlighted the text written by the reviewer in black, and our response in blue. The new version of the modified manuscript where added parts are highlighted in blue have been submitted.
We hope you will find these changes and our response to the reviewer´s comments satisfactory.
We look forward to hearing from you.
With kind regards,
Valentina Casella and Andreas Meyerhans
Reviewer 3
Comments and Suggestions for Authors
In this article by Casella et al. authors tested whether anti-PDL1 immunotherapy affects the fibrosis associated with chronic LCMV infection in a mouse model. This is an interesting study although some of the conclusions are not readily evident from the data presented here. My main suggestions are below.
The title is confusing. Authors should state the main finding of their study rather than providing such a general and confusing statement.
We thank the reviewer for his comment. The title of the manuscript is now changed so that it is clearer and more consistent with the presented data.
The control arm should be included in the scheme.
We now modified the scheme in fig.1a according to the suggestion of the reviewer. We also added a phrase in the figure legend.
Figure 1b: ...”from 0% to 4%, respectively.” Where is the 4% coming from?
Fig.1b shows that the long late anti-PDL1 treatment regimen successfully increases the frequency of LCMV-specific IFNγ-producing CD8 T cells from 0 to approximately 4% of the total CD8 T cells (see data points of day 32).
An improvement in CD8 functionality is not shown in this figure, but rather the development of anti-LCMV responses compared to the control.
The development of the CD8 T cell response during natural, untreated chronic infection is given by the “ctr” data points in Fig.1b. Improvements of the CD8 T cell responses after anti-PDL1 treatments are given by the increase of the responses relative to the controls. For example, improvements from approximately 1.8% to 4% and from 0% to 4% are visible for the short late and long late treatment regimen, respectively, as explained in the text to figure 1b.
To show this, authors should perform longitudinal comparisons: d18 with d30, 29 with 35. How were the responses at these second time points? Figure 1d: “This improved T cell functionality correlated with the decrease in splenic virus loads (Fig.1d)”. This figure does not corroborate this statement. A longitudinal comparison of viral load is required in addition to a correlation test.
Indeed, the time points of measuring CD8 T cell functionality and virus load control were different and thus the correlation between both parameters is only suggestive. Of note, the different time points were chosen by purpose. It was a compromise to observe an optimal functionality increase (day 1 after the last anti-PDL1 treatment) and measure virus load control and fibrosis development, the latter which occurs always after a prolonged immunological insult. To provide direct evidence for the correlation of CD8 T cell functionality and virus load control, we have now included the results of an additional experiment that measured both parameters at the same time (day 60). The data are now shown in figure 2. Anti-PDL1 treatment increased T cell functionality. It was visible even 1 month after treatment termination and linked to virus control. The new data are now described in the result section. Paula Cebollada Rica, the student that participated in that work, was added as a co-authors.
Why was the phenotype of exhausted cells defined as CD44+PD1+Tim3+? Where there any differences when only PD1 was analyzed?
Exhausted CD8 T cells are comprised of cells in distinct differentiation stages, from precursor exhausted T cells to terminal exhausted T cells (see reference 15; Beltra et al., 2020, Immunity). Cells expressing CD44+PD1+Tim3+ are in later stages of exhaustion (DOI: https://doi.org/10.1073/pnas.1009731107) while PD1 expression alone is conventionally used as a marker of T cell activation and thus not sufficient to define exhausted T cells (DOI: https://doi.org/10.1016/j.molcel.2020.02.013). Therefore, a respective analysis of only PD1-expressing cells was not performed.
Line 126: ”Second, from the total pool of exhausted activated cells (Fig. 1c), only a fraction can be functionally reactivated (Fig. 1b)”. Did the authors analyze the frequency of IFN-g+ among the exhausted cells?
We were unprecise with our statement. Only a single immunodominant peptide from LCMV was used for T cell stimulation, and thus only a fraction of the total exhausted T cell pool can be activated under these conditions. Importantly however, this fraction was approximately constant albeit the total frequency of exhausted T cells diminished over time (Fig. 1c). We now modified the respective sentence in the discussion to be more precise.
Concerning the quantification of IFNγ+ amongst exhausted CD8 T cells, we did not perform this analysis because this aspect has already been studied and published in reference 15 (Beltra et al., 2020, Immunity).

Reviewer 4 Report
Comments and Suggestions for Authors
The authors made significant improvements to the manuscript, addressing critical aspects effectively.
Author Response
We like to thank the reviewer for his time to revise our manuscript. There were no further comments.
Round 2
Reviewer 1 Report
Comments and Suggestions for Authors
I find the amended manuscript speculative. The title should read:
Anti-PDL1 checkpoint inhibitor treatment during a chronic virus infection might act below a tissue pathogenicity threshold.
A negative result (the absence of fibrosis) does not allow the authors to abound in speculations (the pathogenicity threshold and chronic therapeutics parallels). In particular, when the experimental design and the presented data are ambiguous and minimal.
Author Response
We thank the reviewer for his thoughtful comment. A modification of the title was also requested by reviewer 3. We have now changed the title of the manuscript so that it is clearer and more consistent with the presented data.
Reviewer 2 Report
Comments and Suggestions for Authors
The manuscript is now well-revised.
Author Response

(The authors gave the same response as above.)
